# Improving uptake of Fracture Prevention drug treatments: a protocol for Development of a consultation intervention (iFraP-D)

Zoe Paskins [1,2] Laurna Bullock,[1] Fay Crawford-Manning [1,2]
Elizabeth Cottrell,[1] Jane Fleming [3,4] Sarah Leyland,[5]
John James Edwards [1] Emma Clark,[6] Simon Thomas,[7]
Stephen Robert Chapman [7] Sarah Ryan,[2,8] J E Lefroy,[1]
Christopher J Gidlow [9] C Iglesias,[10,11] Joanne Protheroe,[1] Robert Horne,[12]
Terence W O'Neill,[13,14] Christian Mallen,[1] Clare Jinks[1]

For numbered affiliations see end of article.

**Correspondence to**
Dr Zoe Paskins;
z.paskins@keele.ac.uk

## ABSTRACT

**Introduction** Prevention of fragility fractures, a source of significant economic and personal burden, is hindered by poor uptake of fracture prevention medicines. Enhancing communication of scientific evidence and elicitation of patient medication-related beliefs has the potential to increase patient commitment to treatment. The *I*mproving uptake of *Fra*cture *P*revention drug treatments (iFraP) programme aims to develop and evaluate a theoretically informed, complex intervention consisting of a computerised web-based decision support tool, training package and information resources, to facilitate informed decision-making about fracture prevention treatment, with a long-term aim of improving informed treatment adherence. This protocol focuses on the iFraP *D*evelopment (iFraP-D) work.

**Methods and analysis** The approach to iFraP-D is informed by the Medical Research Council complex intervention development and evaluation framework and the three-step implementation of change model. The context for the study is UK fracture liaison services (FLS), which enact secondary fracture prevention. An evidence synthesis of clinical guidelines and Delphi exercise will be conducted to identify content for the intervention. Focus groups with patients, FLS clinicians and general practitioners and a usual care survey will facilitate understanding of current practice, and investigate barriers and facilitators to change. Design of the iFraP intervention will be informed by decision aid development standards and theories of implementation, behaviour change, acceptability and medicines adherence. The principles of co-design will underpin all elements of the study through a dedicated iFraP community of practice including key stakeholders and patient advisory groups. In-practice testing of the prototype intervention will inform revisions ready for further testing in a subsequent pilot and feasibility randomised trial.

**Ethics and dissemination** Ethical approval was obtained from North West—Greater Manchester West Research Ethics Committee (19/NW/0559). Dissemination and knowledge mobilisation will be facilitated through national bodies and networks, publications and presentations.

**Trial registration number** researchregistry5041.

### Strengths and limitations of this study

► A robust intervention development process will incorporate multiple sources of evidence, informed by the Medical Research Council framework for developing complex interventions and a three-step model of change.

► A comprehensive logic model and use of the necessity concerns framework provides a theoretical basis for enhancing informed adherence.

► Collaboration with patients and clinicians, using the principles of co-design, and use of the theoretical framework of acceptability to analyse qualitative data will enable us to iteratively develop an intervention that is relevant and acceptable to users.

► *I*mproving uptake of *Fra*cture *P*revention drug treatments will be designed to address barriers to implementation from the outset, through the use of normalisation process theory, theoretical domains framework and the Capabilities, Opportunities and Motivation Behaviour-Based Theory of Change Model (COM-B), and will produce evidence on barriers and facilitators to implementation.

► The research will be conducted in the UK, with in-practice testing conducted at one site; it is possible that barriers and facilitators to change, and the relevance of our intervention may vary across FLS sites, different contexts (eg, in primary care) and geographical locations (nationally and internationally).

## INTRODUCTION

In the UK, 3 million people are estimated to have osteoporosis,[1] contributing to over 500 000 fragility fractures (fractures resulting from low trauma) per year, costing an

estimated £4.4 billion per annum.[2] Fragility fractures can be devastating, sometimes resulting in loss of independence and mortality.[3] Hip fractures alone account for 85 000 unplanned hospital admissions and 1.8 million bed-days in the UK per year.[4] Evidence-based treatments, such as bisphosphonates, are recommended by the National Institute for Health and Care Excellence (NICE) for patients with osteoporosis and/or a high fracture risk. They are inexpensive, cost-effective, readily available and reduce fracture risk by 20%–70% (depending on fracture site).[5] Despite this, a treatment gap exists. Up to 80% of patients who experience a fragility fracture do not receive medication in the year following fracture,[6] 25% of people who are offered medication decline it (non-initiation)[7] and among those who do start bisphosphonates, few persist for long enough for it to be effective, with adherence estimated at 16%–60% at 1 year.[8] Closing this treatment gap may prevent at least 20 000 hip fractures annually in the UK.[4]

Patient reasons for non-initiation and non-persistence of oral bisphosphonates (the mainstay of osteoporosis treatment) are complex and include: perceptions that drugs are not effective, not necessary and/or not safe; limited understanding of the consequences of non-treatment and concerns about perceived or experienced side effects.[9 10] Despite national osteoporosis guidance recommending the provision of information as a core component of management,[11] patients report that osteoporosis information provided in consultations is often not easy to understand.[12] Some primary care clinicians believe that bisphosphonates are not safe, effective or necessary.[13] The more recent shift to base treatment recommendations on fracture risk rather than bone density readings,[14] is not without challenge: patients struggle to understand fracture risk assessments[15] and frequently underestimate their risk of fracture.[16] This suggests unmet health literacy needs and patients have identified improving access to information from health professionals as the number one patient priority for osteoporosis research.[17]

Patients ultimately decide whether to start and continue taking medication, but this decision-making is influenced by the clinician-patient interaction. In order to decide to start and persist with medication, patients need to believe that recommended drug treatment is necessary, relevant, safe and practicable. Effective communication that enables patients to understand complex medical terms and concepts in lay terms, increases patient satisfaction, facilitates participation in the consultation, promotes trust[18] and may increase patients' commitment to medication.[19] This highlights the relevance of promoting and supporting effective communication between clinician and patients, and suggests that improving communication of the harms and benefits of osteoporosis medications may be beneficial in reducing the treatment gap.

Decision aids (DAs) include numerical estimates of risk/benefit. They can facilitate improved risk communication and support patient decision-making before or during healthcare consultations.[20 21] When used across a range of conditions, DAs can increase patient knowledge, reduce decisional conflict, increase patient participation in decision-making and improve the accuracy of risk perception.[20] A recent Cochrane review reported high-quality evidence that DAs, across a range of conditions, increase patient knowledge and reduce decisional conflict, and moderate-quality evidence that DAs increase patient participation in decision-making and improve the accuracy of risk perception.[20] Increases in knowledge and informed choice were also reported in studies where health literacy needs were addressed.[22] Evidence from pooled analyses of studies where there was no equipoise (meaning that DAs were used to give information about recommended drug treatments, rather than to choose between treatment options with perceived equal benefits), has indicated that DAs improve treatment initiation rates.[20] In osteoporosis, use of DAs can increase accuracy of risk perception and shared decision-making. However, existing osteoporosis DAs fail to comprehensively meet international quality standards and patient needs,[23] underlining the requirement for further development.

DAs are sometimes called decision support tools (DSTs), particularly when tools also contain clinician decision support. DSTs can also support clinicians to follow evidence-based practice and have been shown to improve adherence to evidence-based guidelines.[24 25]

The *I*mproving uptake of *Fra*cture *P*revention drug treatments (iFraP) programme aims to develop and evaluate a theoretically informed, complex intervention consisting of a computerised (C)DST, training package and information resources to facilitate informed decision-making about fracture prevention treatment, with a long-term aim of improving informed treatment adherence. This protocol is for the first study within the iFraP programme of work focusing on the *D*evelopment of the new consultation intervention, referred to as iFraP-D.

## STUDY PROTOCOL
### Aims and objectives
The iFraP-D has three overarching objectives, which are to:
1. develop core content for a new consultation intervention (iFraP) based on theory, published guidance, systematic review evidence, a Delphi consensus exercise and stakeholder engagement;
2. design a prototype CDST using published guidance and with stakeholder engagement;
3. use qualitative methods and stakeholder engagement to:
   a. investigate current practice, and the barriers and facilitators to, communicating fracture risk and treatment benefits/harms and to delivering the iFraP intervention (CDST and training package), including the training needs of clinicians;
   b. co-design the components of the iFraP intervention and associated training package for clinicians;

c. plan the integration of components in the new iFraP consultation intervention;

d. conduct cycles of in-practice testing to determine if the iFraP intervention is meeting its objectives, with subsequent refinement of the intervention.

## Overview of context and intervention

The proposed iFraP consultation intervention will be primarily designed for fracture liaison services (FLSs). These are usually nurse-led, and address secondary fracture prevention by: systematically identifying patients with fragility fractures; assessing the patient's bone health, risk of falls and future fracture and providing treatment recommendations to the patient and primary care. Clinical standards for FLS also recommend follow-up consultations, 3 and 12 months postfracture, to support medicines adherence.[26] Within the context of FLS, all consultations are concerned with future fracture prevention, which will maximise efficiency of recruitment and testing. The multidisciplinary study team will allow the relevance of the intervention for use in other primary and secondary care settings in which fracture prevention treatments are recommended to patients, to be considered.

The iFraP consultation intervention will support clinicians to:

► Make decisions about who is eligible for fracture prevention treatment, using a CDST to operationalise existing clinical guidelines.

► In patients in whom fracture prevention treatment is indicated:
  – Communicate the risks and benefits of fracture prevention treatment, including individualised fracture risk, using a CDST, adopting universal precautions for low health literacy.
  – Elicit patients' understanding, and concerns about fracture prevention treatment.
  – Assess readiness to initiate fracture prevention treatments and facilitate behaviour change.
  – Communicate consultation outcomes with the patients' primary care provider and facilitate consistent information provision across primary and secondary care.

The iFraP consultation intervention will be delivered within FLS consultations and will include:

► A CDST to communicate individual fracture risk. This will include clinician decision-support and a patient-facing DA. It will be dynamic, interactive and tailored to risks and needs of the patient. It will incorporate fracture risk (calculated in external systems (eg, FRAX)), an indicator for clinicians of whether treatment is recommended, a pictorial presentation of individualised fracture risk, fracture risk with medication (to show benefits of treatment) and possible treatment harms. The CDST will be used by trained clinicians in a model (face-to-face or remote) consultation with patients.

► Clinician training in delivering the consultation intervention, and supporting delivery of follow-up consultations. This will encompass a prioritised list of key tasks for the clinician (both information giving and eliciting) to undertake. The training will include face-to-face sessions and an e-learning package to introduce the intervention, coach clinicians in listening skills, shared decision-making skills and universal precautions for health literacy, and provide opportunities to practice using the CDST.

► Information resources (paper and online) for the patient and general practitioner to refer to after the initial or follow-up consultation, including a printout from the CDST.

## Theoretical frameworks underpinning iFraP development

The long-term aim of the iFraP intervention is to improve fracture prevention treatment uptake, thereby preventing future fragility fractures. The intervention is targeted for use by clinicians together with the patient. This will empower clinicians to support patient behaviour change, and support clinicians with the skills to elicit and address patient perceptions and practicalities related to treatment adherence. The necessity-concerns framework (NCF) will be used as an overarching theoretical framework when designing the iFraP intervention to understand patient's attitudes and beliefs underpinning treatment non-adherence.[27] The NCF suggests that medication adherence is influenced by implicit judgements of personal need for the treatment (necessity beliefs) and concerns about the potential consequences of treatment.[27]

To optimise iFraP in terms of acceptability to clinicians and patients, we will use the theoretical framework of acceptability (TFA)[28] as an overarching framework to inform iFraP intervention design. The TFA consists of seven constructs of acceptability of healthcare interventions (affective attitude, burden, intervention coherence, ethicality, perceived expectations, opportunity cost and self-efficacy). These constructs will inform data collection and analysis.

The implementation of iFraP is informed by the Capabilities, Opportunities and Motivation Behaviour-Based Theory of Change Model (COM-B)[29] and the complementary theoretical domains famework (TDF),[30] which aim to simplify and integrate a range of behaviour change theories. The TDF identifies 14 related domains of influence on behaviour. In iFraP-D, data collection will be informed by domains within the TDF (eg, knowledge, beliefs about capabilities, skills and goals). This will identify potential barriers and enablers for clinicians to implement iFraP ahead of a formal feasibility study. normalisation process theory (NPT)[31] will underpin the investigation of the dynamics of implementing, embedding and integrating iFraP, in order to identify potential process problems related to implementing iFraP in the next phase of feasibility testing. NPT ensures consideration of potential structural problems about the integration of iFraP into existing services.

The specific programme theory for the iFraP-D study is detailed in the study logic model (figure 1). This initial

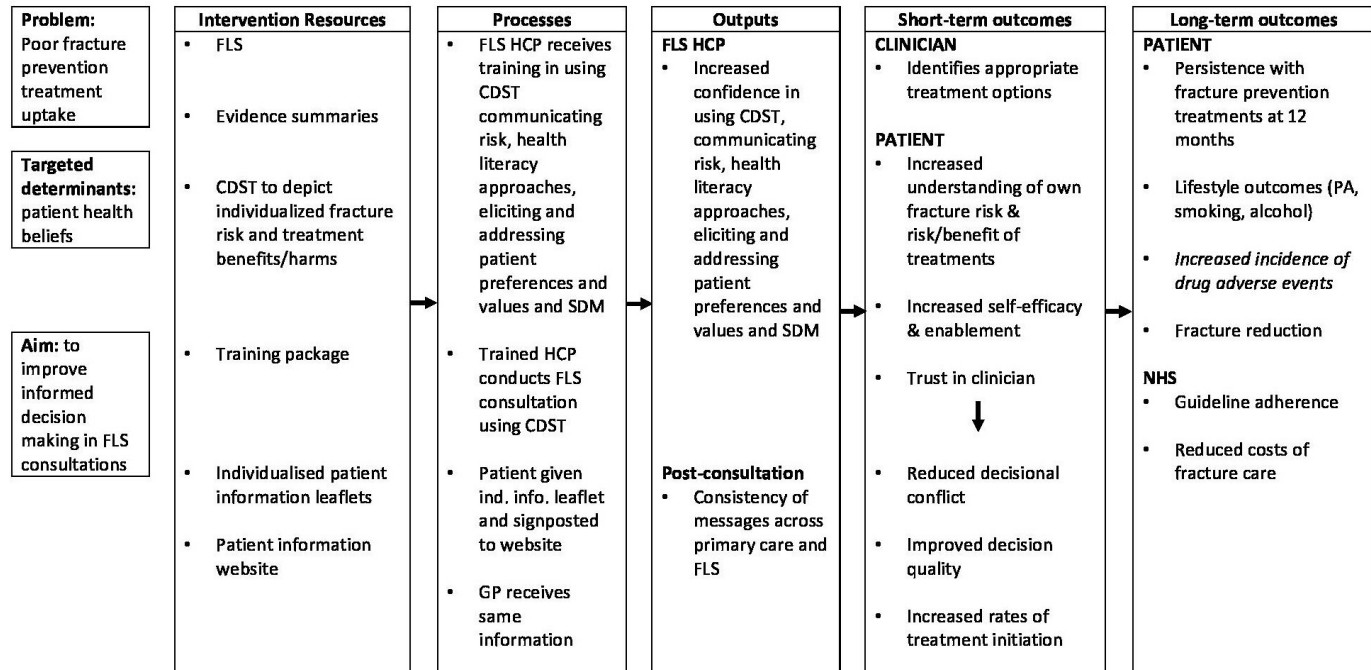

**Figure 1** *I*mproving uptake of *F*racture *P*revention drug treatments *D*evelopment logic model. Context: consultations conducted in pre-existing specialist face-to-face fracture prevention services (FLS). Contexual factors: poor uptake of the National Institute for Health and Care Excellence/National Osteoporosis Guideline Group guidelines; disconnect in advice given to patient between FLS and primary care; media and wider social influences on health behaviours. CDST, computerised decision support tool; FLS, fracture liaison service; GP, general practitioner; HCP, healthcare professional; NHS, National Health Service; PA, physical activity; SDM, shared decision-making.

logic model may undergo iterative changes throughout the iFraP programme. Throughout development, a health economist will define the decision problem enabling identification of the different elements of treatment effect potentially associated with the iFraP intervention and directly informing the data collection strategy, including primary and secondary outcomes, for the subsequent pilot and feasibility trial that will follow iFraP-D.

### Approach to iFraP intervention development

In line with key principles of intervention development, our approach will be iterative, open to change and forward looking to future evaluation and implementation.[32] The Medical Research Council guidance on complex intervention development will be used as an overall framework,[33] within which we will specifically use a three-step implementation of change model.[34] This approach incorporates three pragmatic questions with three steps, each linked to iFraP study objectives outlined previously:

▶ *Step 1*: "Where do we want to be?" Make a concrete proposal for change, and develop the content and format of the consultation intervention (objective 1).
▶ *Step 2*: "Where are we now?" Understand the current clinical context including barriers and facilitators to change (objective 3a).
▶ *Step 3*: "How do we get there" Develop a strategy to change behaviours, by designing and refining the prototype into a draft intervention, and field-testing (objective 2, 3b, 3c and 3d).

To answer the above questions, we will take actions common to different intervention development approaches;[35 36] drawing on theory, empirical evidence from the evidence synthesis, Delphi survey, qualitative focus groups and in-practice testing, stakeholder engagement and guidance for the development of DAs.[37 38] We will adopt an informed design to iFraP design decision-making.[32 39] We will frequently engage with stakeholders as part of our community of practice (CoP) and patient advisory groups (PAG) to discuss ideas generated by the study team and research findings.[39] The informed design will facilitate stakeholder involvement in iFraP intervention content and design decisions,[39] with final design decision-making made and documented by the Study Management Group[32] supported by the APEASE criteria (Affordability, Practicality, Effectiveness, Acceptability, Side-effects/safety, and Equity) where appropriate.[40]

Each step and linked study objective are outlined in turn throughout this protocol. The inter-relation between the three steps and methods of data collection is displayed in figure 2.

### Co-design of the iFraP intervention
#### Community of practice

CoPs bring together expertise with a common concern or interest, with the aim of improving and learning to do better through regular group interaction.[41] We will bring together relevant expertise (FLS clinicians, GPs, osteoporosis specialists, commissioning representatives,

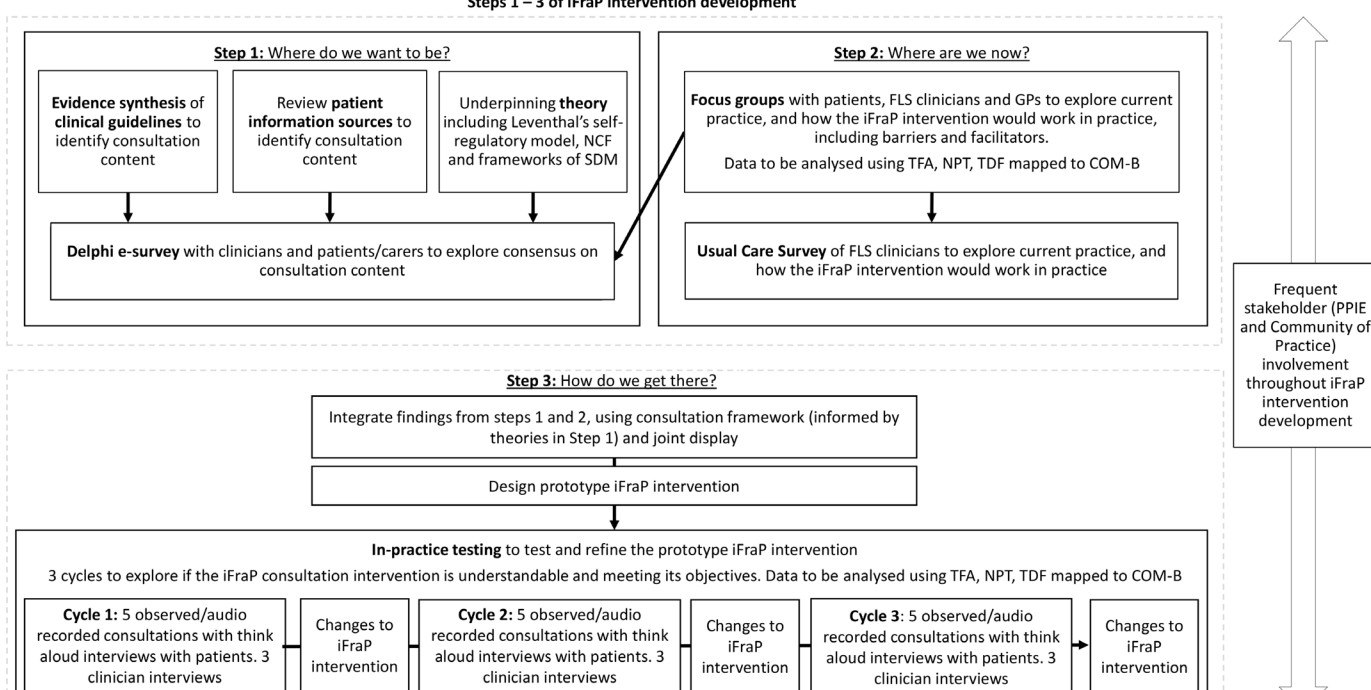

**Figure 2** Inter-relation between the implementation of change model and *I*mproving uptake of *Fra*cture *P*revention drug treatments *D*evelopment (iFraP-D) data collection methods. COM-B, Capabilities, Opportunities and Motivation Behaviour-Based Theory of Change Model; FLS, fracture liaison service; GP, general practitioner; NCF, necessity concerns framework; NPT, normalisation process theory; PPIE, patient and public engagement and involvement; SDM, shared decision-making; TDF, theoretical domains framework; TFA, theoretical framework of acceptability.

patients (with experience of fracture prevention treatment, supported by a patient and public engagement and involvement (PPIE) worker)), representatives from the Royal Osteoporosis Society (ROS) and Health Literacy UK and a behaviour change expert) in a stakeholder group that adopts a CoP approach.[41] The CoP will provide their views on the iFraP intervention content, and discuss the findings from the evidence synthesis, Delphi study, qualitative focus groups and in-practice testing.

### Patient and public involvement and engagement

The osteoporosis Research User Group (RUG) at Keele University comprises patients with experience of osteoporosis and/or fragility fractures, or carers for such patients. These RUG members had substantial involvement in a previous study to identify patient and public priorities for research in osteoporosis, which provided the starting point for iFraP.[17] Furthermore, the study-specific PAG informed and agreed how PPIE members will be involved throughout the iFraP programme at the outset. In addition to PAG members participating in the Study Management Group, Award Steering Committee meetings and CoP meetings, approximately six study-specific PAG meetings will be convened throughout the iFraP research cycle. PAG meetings will: (i) facilitate design, analysis and interpretation of the Delphi study (including how best to recruit and explain the Delphi study to patients, piloting the survey and contributing to analysis), qualitative focus groups and in-practice testing (by informing recruitment,

co-designing topic guide content and supporting analysis and interpretation of findings); (ii) co-design visual components of the CDST; (iii) informally test the CDST prototype before in-practice testing and (iv) provide advice about dissemination to the wider public, including what results to share, when and in what format. Additional information regarding PPIE is detailed throughout this protocol where appropriate. The Guidance for Reporting Involvement of Patients and the Public (GRIPP2) will be used in future dissemination to guide PPIE reporting.[42]

## METHODS AND ANALYSIS

The iFraP-D study began at the end of March 2019 and the study is expected to be completed in December 2021.

### Step 1: developing content and format of iFraP

In order to inform the content of the model consultation and the CDST components (overarching objective 1) of the iFraP consultation intervention, an evidence synthesis will be conducted of clinical guidelines, followed by consultation with stakeholders and a Delphi study with patients/carers and clinicians.

### Evidence synthesis and stakeholder consultation
#### Search strategy

To identify osteoporosis clinical guidelines, the NHS Evidence electronic database will be searched using keywords 'fragility fracture' and 'osteoporosis' and

'guidelines' for studies that fulfilled the eligibility criteria (online supplemental table 1). The search will be filtered to include guidance, quality indicators and policy and strategy. Guidelines that were developed, reviewed or revised within the 10 years prior to search inception will be used. Generic NICE guidance relating to conducting the consultation will also be included.

Patient information resources will be identified using a Google search. Leaflets and webpages will be selected because they are easily available (eg, on the internet, through patient organisations or places that people with osteoporosis might visit such as pharmacies). They will come from three different types of source: healthcare providers (UK National Health Service), charitable and voluntary organisations and the medical profession.

### Selection process

Eligible guidelines will be selected on title first by one reviewer (ZP). Full texts will be retrieved and assessed if the title, summary or abstract provides insufficient information.

Eligible patient information resources will be selected by two reviewers. A pragmatic and purposive sample of resources that are most commonly accessed and represent the three groups of information providers will be selected.[43]

### Quality appraisal

The quality of the included guidelines will be appraised using the Appraisal of Guidelines for Research and Evaluation (AGREE II) tool.[44] Guidelines that score 75% or above will be tagged as high quality (in line with examples given by the AGREE II developers); this quality score will inform discussion in the stakeholder groups about the relevance of recommendations.

The overall quality of each patient information resource will be assessed using a modified International Patient Decision Aid Standards (m-IPDAS).[37] The m-IPDAS assess patient information based on 32 items across 8 categories of content, bias, detail, probabilities, accuracy, decision-making, conflict of interest, structure and layout and reliability.

### Data extraction

Statements and recommendations from included guidelines that are relevant to tasks for the clinician in the consultation will be extracted and grouped into steps of phases of the consultation. Text from patient information resources will be extracted relating to descriptions of osteoporosis and osteoporosis drug treatment using a bespoke data extraction form in excel.

### Analysis

A narrative summary will report the findings of the guideline evidence synthesis, including textual description of guidelines, tabulation of recommendations and exploration of relationships between and within guidelines.

Extracted data from the patient information will be thematically analysed independently by two members of the study team using the five domains of Leventhal's Common-Sense Model of Disease as a deductive framework.[45] Discussions will check the consistency of coding against the framework.

### Stakeholder consultation

Evidence synthesis findings will be presented to the CoP for reflection on the findings and discussion of any inconsistencies, and discussion of how clinician tasks map to stages of the FLS consultation. The synthesis output (informed by the CoP) will form the basis of the Delphi.

### Delphi study
#### Design

Three rounds of a modified Delphi consensus study[46] will be conducted. The Delphi approach is described as 'modified', as participants will be presented with a list of statements to consider rather than generating their own list. However, in the first round, they will also have the opportunity to suggest additional statements for the consultation in free-text boxes. The aim of the Delphi is to make decisions about inclusion and exclusion and for included statements to ascertain essential and optional consultation content. The same survey will be developed for all participants (both clinical and patient/carer) to develop a common, understandable language. Survey participants will be presented with clinical vignettes and a list of potential content for the model consultation. The content will be derived from the evidence synthesis, informed by the CoP, and written in partnership with PAG members.

#### Participants

Patients with osteoporosis and/or fragility fractures and/or their carers will be recruited through the ROS supporters' network. Clinicians who have experience of consulting with patients, where fracture risk is calculated and fracture prevention drug treatments are recommended, will be invited to participate through ROS clinician mailing lists and the study team's clinical networks. Clinicians will be multidisciplinary; from primary and secondary care and academic settings. We aim to invite up to 400 participants, anticipating a 40% initial response rate and subsequent drop out at rounds 2 and 3, with the aim of achieving a final sample of 15 patients/carers and 15 clinicians.

#### Data collection and analysis

In round 1, statements will be presented that relate to tasks of the consultation, to include clinician decision-making tasks and considerations, eliciting information, giving information, example explanations and hypothetical use of the CDST, for example, 'the patient should be verbally informed of their individualised fracture risk', and 'the patient should be shown a picture of their individualised fracture risk'. Statements will be grouped together under overarching headings (eg, 'share

information about condition' and 'signpost next steps') that reflect stages of the consultation (as discussed with the CoP). The participant will be asked to rate their perception of the importance of each statement on a 5-point Likert scale.

Patient/Carer and clinician responses will be analysed separately. Mean scores will be calculated. Items with a mean importance score of <4 (range 1–5) in both patient and clinician surveys will be removed. Items scoring a mean score of <4 in either patient/carer or clinician survey will be individually reviewed by the Study Management Group (including PPIE members) to make a decision about whether the item progresses to round 2, informed by underpinning theory, other emergent findings in iFraP-D (eg, focus groups), PPIE member experiences and views, etc.

In round 2, newly suggested statements in free-text comments, along with emerging findings from iFraP-D (eg, focus group findings, stakeholder (CoP and PPIE) experiences and views) will be added to statements progressing from round 1. Participants will be shown mean scores of importance from round 1 and asked to re-rate the importance of each statement, again using the same 5-point numeric Likert scale.

After round 2, statements will be ranked according to their mean rating score (calculated as per round 1). The top scoring statements from both patient/carer and clinician surveys will progress to round 3, identified as those in the top three of four quartiles. Statements with a patient/carer and/or clinician score in the lowest quartile will be reviewed by the Study Management Group (including PPIE members), as to whether the statement should progress to round 3, informed by the considerations described in round 1.

In round 3, low ranking items will be removed and participants will be asked to agree/disagree whether a statement is essential in a time-limited consultation, or not. Percentage agreement that 'this item is essential in the time-limited consultation' will be calculated. Statements will be ranked according to percentage agreement for both patient/carer and clinician surveys. To identify a set number of essential consultation statements, which can realistically be undertaken in a time-limited consultation, we will not predefine an arbitrary level of agreement for a task to be included in the iFraP consultation. The Study Management Group in conjunction with PPIE members and expert advisors will review the ranked findings and identify, using their own knowledge and experience and free-text responses, where the most appropriate threshold is for tasks or considerations to be classified as 'essential' or 'optional'.

### Step 2: to understand the current clinical context of FLS consultations including barriers and facilitators to change

This step will involve mixed methods of data collection. A qualitative focus group study and a survey of usual care will achieve objective 3a.

### Focus groups
#### Objective
To qualitatively explore current practice in relation to fracture prevention communication, anticipated barriers and facilitators to the iFraP intervention and perceived training needs for FLS clinicians.

#### Participants
Two focus groups with patients that have recently attended an FLS consultation and received a treatment recommendation, two focus groups with clinicians that conduct face-to-face consultations with patients in UK FLSs and one focus group with GPs with experience of consulting with patients who have been seen in FLS and that work in the catchment of a UK FLS. Focus groups will be conducted with participants from different geographical areas, with approximately four to eight participants in each group.

#### Data collection
Focus groups enable investigation of practices among individuals and are particularly suited to studying group norms and processes, as group interaction is explicitly used to generate data and insights.[47] If face-to-face focus groups are inappropriate, qualitative data collection will be conducted remotely (eg, by telephone or video conferencing software) and audio-recorded. A topic guide (informed by theory, evidence gathering and CoP/PPIE views) will prompt, facilitate and structure discussion. The guide will include questions related to the TDF as this framework will inform the content of training to enable skills development for clinicians and is therefore relevant to understanding how clinicians would use iFraP. The questions will also be informed by the TFA; our overarching TFA enabling the investigation of seven constructs of acceptability. Topic guides will be iteratively developed during the study to account for insights not anticipated.

#### Analysis
Focus groups will be audio-recorded and transcribed. We will use a framework approach to analysis,[48] including a two-stage process to (1) inductively code transcripts followed by (2) a deductive exercise to map identified codes to the domains of relevant theoretical frameworks, including the TDF, TFA, NPT, in accordance with previous research.[49 50] Theoretical frameworks will provide a lens to deepen our understanding of current FLS practice, intervention acceptability and barriers to, and facilitators of intervention implementation. For example, TDF domains will be mapped to the COM-B model to identify intervention functions and theory-driven behaviour change techniques.[40] Analysis will be undertaken by two members of the study team independently, with a mapping exercise to explore coding consistency. Findings will be discussed with the wider study team and PAG members. Qualitative data will be managed and coded using NVivo V.11.

### Usual care survey
*Participants*

An electronic survey (using HealthSurvey, hosted by Keele University) will be designed and distributed to nurses or allied health professionals working in UK FLSs, with a target sample size of 80 across the 4 devolved nations. The survey will be distributed by the Royal College of Physicians Fracture Liaison database national audit, ROS mailing lists and by researchers to individuals in their known professional networks.

*Data collection*

Variation in FLS design and specification influences the amount of clinician-patient dialogue about drug recommendations, which has direct implications for implementation of the iFraP intervention and any subsequent trial. For example, the extent of patient contact (eg, face-to-face, telephone or by letter) will affect the ability to use the CDST within the consultation. The specific content for the usual care survey will be informed by the focus groups, and will aim to describe and quantify the nature and amount of discussion FLSs have with patients about osteoporotic drugs. Any planned changes in service specification will also be explored.

*Analysis*

Any duplicate entries from the same FLS will be checked for consistency. If there are any differences, the respondent will be emailed to clarify. A descriptive analysis will be undertaken. Findings will be discussed with the CoP to discuss implications for iFraP intervention design.

Results of the descriptive analysis will also be used to create a detailed FLS typology, based on the extent of face-to-face contact to facilitate sampling for the subsequent pilot and feasibility trial.

### Step 3: design and refine the iFraP intervention using field testing

The outputs from studies in step 1 and step 2 will be integrated to inform design of the CDST and training (objective 2, 3b, 3c). iFraP design decision-making will be made and documented by the Study Management Group.[32] Following design of the prototype iFraP consultation intervention (including CDST and training), we will conduct three cycles of in-practice (or field) testing to assess how the fracture prevention components work together, and the acceptability and feasibility of the prototype CDST (objective 3d).

### Data integration

Data from the evidence synthesis, Delphi survey and focus groups will be integrated to add rigour, provide greater credibility to results and generate insights about the intervention in development. Integration will be achieved using a convergent design and merging of the quantitative and qualitative findings using side-by-side joint displays.[51]

A framework for the consultation will be drafted, informed by the consultation stages identified in the evidence synthesis, underpinning theory, frameworks of shared decision-making and principles of health literacy. This will include a series of stages and tasks for the clinician to progress through during the consultation, including gathering information, clinician decision-making, eliciting patient knowledge through to summarising and signposting. Findings from the Delphi analysis (a list of items to be included—and whether they are essential or optional, a list of items excluded, and relevant free-text comments) and qualitative analysis will be mapped to the relevant stage of the consultation in the drafted framework. A separate extraction table will be used for joint display of other contextual considerations for the intervention, derived from the free-text in the Delphi and qualitative findings, and will be organised into considerations with relevance to (i) the tool development, (ii) the training development and/or (iii) subsequent trial design.

The joint display will be interrogated to identify 'meta-inferences':[51] confirmed, discordant and expanded findings. Resultant discordant areas, or areas of uncertainty, will be presented to the CoP and PAG for further discussion. Following stakeholder discussion, the list of content and considerations for (i) tool development, (ii) training development and (iii) the trial will be finalised.

### iFraP intervention design
*Prototype CDST design*

The CDST will contain content to support clinician and patient decision-making. The patient-facing component will be designed with the principles of conversation aids, rather than DAs, to support the discussion between clinician and patient.[52] Frequent stakeholder (CoP and PPIE) workshops throughout iFraP-D will allow members to contribute and provide insight into CDST design. Insights will be considered alongside integration outputs described above in accordance with our informed design approach.[32 39] The prototype tool will be web-based and built using a programmed decision-tree based on a modified Markov model (a way to model prognosis for clinical problems with ongoing risk). Its development will adhere to international guidance and be informed by a published and process map for DST development.[38] A technical production group will manage the CDST design. A scientific advisory group of osteoporosis academics will advise on the scientific evidence-base that will underpin the CDST. Where possible, event rates from existing NICE guidelines and meta-analyses will be used. PPIE and clinician members of the CoP will informally test early CDST prototypes.

*Training design*

Separately, also informed by stakeholder (CoP and PPIE) workshops and integration outputs (eg, list of content and considerations for iFraP development), a training development group (including input from a behavioural psychologist, patients, expert nurse educator and expert in shared decision-making) will map behaviour change techniques targeted to FLS clinician training needs. Qualitative analysis mapped to the TDF[30] aligned with the COM-B model will support identification of theory-driven behaviour change techniques.[29] This process will allow the training

development group to develop an evidence-based clinician training package and associated training manual.

## In-practice testing
### Design
We will conduct three cycles of in-practice testing of the prototype iFraP consultation in one FLS site. Each of the three cycles will consist of five consultations with patients and postconsultation interviews. At the end of each cycle, the FLS clinician conducting the prototype iFraP consultation will also be interviewed. Each consultation will be audio-recorded, observed and contain a 'think-aloud' interview

### Participants
Patients with a recent fragility fracture (n=15) who are referred for an FLS consultation and clinicians conducting face-to-face FLS consultations (up to n=3). In-practice testing will take place at the Haywood Hospital FLS chosen for its geographical proximity to the study team. The Haywood Hospital FLS operates a one-stop model, meaning that if appropriate, patients have a bone density scan (dual-energy X-ray absorptiometry), nurse assessment, drug treatment recommendation and blood tests as part of one consultation.

### Data collection and procedures
Patients with FLS appointments will be asked to consent to having their consultation observed by a researcher and audio-recorded. For each patient participant, the researcher will conduct a brief (previously developed) warm-up exercise on the 'think-aloud' technique prior to the consultation.[53] In each cycle of consultation and 'think aloud' interview, a three-step test interview approach will be used.[54] The topic guide will be developed in collaboration with PAG members.

First, the researcher will observe and listen to the test consultation (noting any visual and verbal cues to be explored in the interview immediately following the consultation) and will make notes using an observation schedule. Patients will be asked to mention (aloud, at the time they arise) any thoughts and feelings, if they feel comfortable to do so, during the consultation as part of a 'think-aloud' process.

Second, patient participants will be interviewed on completion of the consultation using predefined and spontaneous probes related to iFraP intervention components, to follow-up issues that emerged during the consultation (eg, to explore unexpected responses and fill in gaps where the participant may not have commented in real-time). This step will provide evidence on whether risk communication is understood by participants in a consistent way and in the way intended, and how well the intervention is meeting its objectives.

Third, participants will be invited to explain comments made in step 2 and their behaviour during the consultation, and share views and opinions of the consultation. They will be encouraged to say everything they thought

about each part of the consultation. Participants will be invited to contact the researcher if any further thoughts come to mind after the interview.

Postconsultation interviews with clinicians will be conducted as soon after the last consultation in the cycle as possible. A similar three-step approach will be used. In the first step however, these clinician participants will be played audio clips demonstrating where they explained risk, used the CDST and gave recommendations. They will be encouraged to think aloud and mention any thoughts or feelings during playback. Second and third steps will be for patient interviews.

### Analysis
The framework approach (as described above for focus groups) will be used. The first interviews will be analysed, with help from CoP and PAG members to interpret data. Relevant changes to iFraP will be made ahead of further testing, supported by theoretical frameworks to identify and overcome barriers to implementation, enabling an iterative cycle moving between user feedback and changes to the intervention.

The outcome of this study will finalise a draft iFraP intervention and intervention manual to be tested in a formal pilot and feasibility trial with nested value of information analysis, to evaluate acceptability, feasibility and cost-effectiveness.

## ETHICS AND DISSEMINATION
Ethics approval for the work outlined in this protocol was sought and obtained from North West—Greater Manchester West Research Ethics Committee (reference number: 19/NW/0559).

Dissemination and knowledge mobilisation will be facilitated through national bodies and networks such as the ROS, journal papers and conference presentations. The results of this study will be made widely and freely available to all stakeholders; a summary of the results will be published on the Keele University and ROS website. PAG members will advise on how to translate these into easily understandable messages and on how best to disseminate the results to the wider public.

In addition to publications in open-access peer-reviewed journals, we will use NHS networks and links to professional bodies to support dissemination of the findings to all stakeholders and will use social media to promote the findings via our dedicated Twitter and Facebook feeds.

**Author affiliations**
[1]School of Medicine, Keele University, Keele, UK
[2]Haywood Academic Rheumatology Centre, Haywood Hospital, Stoke-on-Trent, UK
[3]Cambridge Public Health, University of Cambridge, Cambridge, UK
[4]Cambridge University Hospitals NHS Trust, Addenbrooke's Hospital, Cambridge, UK
[5]Royal Osteoporosis Society, Bath, UK
[6]Bristol Medical School, University of Bristol, Bristol, UK
[7]School of Pharmacy and Bioengineering, Keele University, Stoke-on-Trent, UK
[8]School of Medicine & School of Nursing and Midwifery, Keele University, Stoke-on-Trent, UK
[9]Centre for Health and Development, Staffordshire University, Stoke-on-Trent, UK

[10]Department of Health Sciences, University of York, York, UK
[11]Danish Centre for Healthcare Improvements, Aalborg Universitet, Aalborg, Denmark
[12]Centre for Behavioural Medicine, UCL School of Pharmacy, University College London, London, UK
[13]Centre for Epidemiology Versus Arthritis, University of Manchester, Manchester, UK
[14]NIHR Manchester Biomedical Research Centre, Manchester University NHS Foundation Trust, Manchester, UK

**Contributors** ZP conceived the study. ZP, LB, ECo, JF, SL, JJE, ECl, ST, SRC, CJG, CI, TWO'N, CM and CJ reviewed the study design and contributed to study implementation as part of the iFraP Study Management Group. JP and RH contributed to study design as expert study advisors. FC-M will support study set up and data collection. JEL, SL and SR will form the iFraP training development group. ST and SRC will inform design and development of the prototype iFraP CDST. CI will provide health economic expertise. All authors contributed to refinement of the study protocol and approved the final manuscript.

**Funding** This study was funded by the National Institute for Health Research (NIHR) (Clinician Scientist Award (CS-2018-18-ST2-010)/NIHR Academy), the Royal Osteoporosis Society (REF: 430) and the Haywood Rheumatology Research and Development Foundation (not applicable).

**Competing interests** ZP reports grants from the NIHR Clinician Scientist Award (CS-2018-18-ST2-010), Royal Osteoporosis Society and the Haywood Rheumatology Research and Development Foundation to conduct this study. CJ reports funding from the NIHR Applied Research Collaboration (ARC) West Midlands; CM reports grants from the NIHR Research Professorship in General Practice (NIHR-RP-2014-04-026), the NIHR School for Primary Care Research and NIHR Applied Research Collaborations (West Midlands), Wellcome, Medical Research Council, Dunhill, Versus Arthritis and Bristol Myer-Squibb, outside the submitted work; FC-M reports grants from NIHR Clinical Research Network Scholar Programme; ST and SRC report receiving funds from National Institute for Health Research (NIHR) (Clinician Scientist Award (CS-2018-18-ST2-010)/NIHR Academy) and the Royal Osteoporosis Society via Prescribing Decision Support Ltd to support the development of the iFraP decision support tool; TWO'N reports grants and non-financial support from AMGEN and the NIHR Manchester Biomedical Research Centre outside the submitted work; CI reports grants from NIHR during the conduct of the study and disclosed being a member of NICE's Medical Technologies Advisory Committee outside the submitted work; JJE reports grants as a NIHR Academic Clinical Lecturer in Primary Care (CL-2016-10-003). ECo reports grants from Versus Arthritis, NIHR RfPB, HQIP and the Royal Osteoporosis Society.

**Patient consent for publication** Not required.

**Provenance and peer review** Not commissioned; externally peer reviewed.

**ORCID iDs**
Zoe Paskins http://orcid.org/0000-0002-7783-2986
Laurna Bullock http://orcid.org/0000-0002-4193-1835
Fay Crawford-Manning http://orcid.org/0000-0002-9768-1695
Elizabeth Cottrell http://orcid.org/0000-0002-5757-1854
Jane Fleming http://orcid.org/0000-0002-8127-2061
John James Edwards http://orcid.org/0000-0003-0432-7783
Stephen Robert Chapman http://orcid.org/0000-0002-0326-7742
Christopher J Gidlow http://orcid.org/0000-0003-4990-4572

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
