## [Reviewer comments · BMJ Open]

ARTICLE DETAILS

TITLE (PROVISIONAL)	Improving uptake of Fracture Prevention drug treatments: a protocol for Development of a consultation intervention (iFraP-D)
AUTHORS	Paskins, Zoe; Bullock, Lurna; Crawford-Manning, Fay; Cottrell, Elizabeth; Fleming, Jane; Leyland, Sarah; Edwards, John; Clark, Emma; Thomas, Simon; Chapman, Stephen; Ryan, Sarah; Lefroy, JE; Gidlow, Christopher; Iglesias, C; Protheroe, Joanne; Horne, Robert; O'Neill, Terence; Mallen, Christian; Jinks, Clare

VERSION 1 – REVIEW

REVIEWER	Roigk, Patrick Robert Bosch Hospital
REVIEW RETURNED	17-Apr-2021

GENERAL COMMENTS	Dear authors, your work is very important and fills a gap within the area of secondary fracture prevention. However, I have some additional comments for you: Page 11 Line 6: 'CDST' please introduce always any abbreviation. Page 11 line 16: please add information about the planned training of the clinicians, if possible. Page 11 Line 21: You will use a wide range of theoretical frameworks and theories. Please bring them together e.g. in a figure to make it more clear in which step of the research process they will be used and how the results of each step will be integrated (maybe within figure 2? or in a timeline?). Figures: please add the description below the figures. Page 15 line 11-35: Please think about changing the order: after you describe round 3, you describe the analysis of the steps. After, you started again with round 2 and later again round 3. I would suggest to end with the analysis. Page 16 Line 17: Do you only record the focus groups by audio? If you will conduct them per by video (line 8-9) i think the program will only aloud to record by video and audio recording. Furthermore, I would suggest to record by audio and video particular when you interview up to 8 persons (line 4).
---

REVIEWER	Close, Jacqueline University of New South Wales
REVIEW RETURNED	09-Jun-2021

GENERAL COMMENTS	The protocol is well written, with sufficient detail and clarity for the reader to understand what is being proposed. I don't have any methodological or ethical questions.
---

	Minor point - could the link to the Research registry be inserted - it took me a while to find it.
--	--

VERSION 1 – AUTHOR RESPONSE

Reviewer 1

Reviewer Comment:

‘CDST’ please introduce always any abbreviation.

Response:

We thank the reviewer for their comment. We introduce the abbreviation ‘DST’ and further introduce ‘computerised (C)DSTs’ towards the end of the introduction:

“DAs are sometimes called Decision Support Tools (DSTs), particularly when tools also contain clinician decision support” ... “complex intervention consisting of a computerised (C)DST, training package and information resources” [see page 6, lines 7-12]

Reviewer Comment:

Please add information about the planned training of the clinicians, if possible.

Response:

Thank you to the reviewer for highlighting the need for additional information about the clinician training. We have added detail to the following explanation:

“This will encompass a prioritised list of key tasks for the clinician (both information giving and eliciting) to undertake. The training will include face-to-face sessions and an e-learning package to introduce the intervention, coach clinicians in listening skills, shared decision-making skills and universal precautions for health literacy, and, provide opportunities to practice using the CDST.” [page 8, lines 16-19]

Reviewer Comment:

You will use a wide range of theoretical frameworks and theories. Please bring them together e.g. in a figure to make it more clear in which step of the research process they will be used and how the results of each step will be integrated (maybe within figure 2? or in a timeline?).

Response:

We thank the reviewer for this helpful comment. We have followed the reviewer’s recommendation to outline which theoretical frameworks and theories will be used, at what stage, and how they will be integrated in Figure 2.

Reviewer Comment:

Figures: please add the description below the figures.

Response:

Figure captions were included in the BMJ Open file upload during the submission process. We also provided figure captions at the end of the manuscript in line with BMJ Open guidelines. All figure captions should be visible in the final version of the manuscript.

Reviewer Comment:

Please think about changing the order: after you describe round 3, you describe the analysis of the steps. After, you started again with round 2 and later again round 3. I would suggest to end with the analysis.

Response:

The manuscript was organised so that details about the Delphi survey data collection at rounds 1, 2 and 3 were ordered sequentially. Following data collection, we followed the same approach by detailing the analysis of each Delphi survey round in turn. We have now amended to discuss data collection and analysis together, for each round. We believe that this structural change has improved the flow of this section.

Reviewer Comment:

Do you only record the focus groups by audio? If you will conduct them per by video (line 8-9) i think the program will only aloud to record by video and audio recording. Furthermore, I would suggest to record by audio and video particular when you interview up to 8 persons (line 4).

Response:

We thank the reviewer for highlighting the need for clarity about audio- and video-recording. We have added detail to emphasise that regardless of in-person or remote data collection, only audio-recordings will be obtained, in line with our ethical approval.

“If face-to-face focus groups are inappropriate, qualitative data collection will be conducted remotely (e.g. by telephone or video conferencing software) and audio-recorded.” [page 13, lines 7-9]

Reviewer 2

Reviewer Comment:

could the link to the Research registry be inserted - it took me a while to find it.

Response:

We thank reviewer 2 for their positive comments and helpful feedback. We have added a link to facilitate access to the research registry:

“Research Registry (identifying number: researchregistry5041)

<https://www.researchregistry.com/browse-the-registry#home/registrationdetails/5d3af8eb65383300105d9e68/>”

VERSION 2 – REVIEW

REVIEWER	Roigk, Patrick Robert Bosch Hospital
REVIEW RETURNED	28-Jul-2021
GENERAL COMMENTS	Dear Authors, your research is very important. After your minor revision of the manuscript, I recommend to accept it.